# Extrapolating Large Models from the Small: Optimal Learning of Scaling Laws

## Abstract

As large language models (LLMs) continue to scale to billions of parameters, training them becomes increasingly expensive, making it infeasible to exhaustively explore the vast design space including model architectures, parameter sizes, and compute budgets. Scaling laws have therefore emerged as an essential tool for predicting the performance of larger models by extrapolating from smaller ones, enabling practitioners to make informed design choices without full-scale training. However, existing approaches lack formal guarantees on the predicted results and overlook the out-of-distribution nature of such extrapolation, leading to instability and unreliable predictions. We address these challenges with three key contributions. First, we introduce *Equivalent Sample Size* (ESS), a principled and interpretable metric that quantifies prediction uncertainty by translating it into the number of test samples required for direct, in-distribution evaluation. Second, we analyze how extrapolation amplifies prediction variance and develop an efficient algorithm that optimally allocates smaller-model evaluations to maximize ESS under compute budgets. Third, experiments on both simulated and real datasets show that ESS and our algorithm guide the design of scaling-law learning, cut evaluation cost, and deliver reliable LLM performance predictions.

## 1 Introduction

The development of large language models (LLMs) has been driven by the intuition that scaling up model size, data volume, and compute generally leads to better performance (Kaplan et al., 2020). However, pre-training modern LLMs requires immense computational resources, often costing millions of dollars and weeks of training time across thousands of GPUs. As a result, blindly increasing one dimension such as model size, has become increasingly inefficient and impractical. There is a growing demand for principled tools that can guide pre-training decisions before these expensive investments are made Touvron et al. (2023); Bi et al. (2024); Choshen et al. (2024).

A promising direction is the use of scaling laws, which posit power-law relationships between model performance and key design variables such as model size and dataset size (Rosenfeld et al., 2019; Kaplan et al., 2020; Hoffmann et al., 2022). These empirical laws enable predicting the performance of a target large model by extrapolating from the observed performance of smaller models – a framework that we referred to as *scaling prediction* through the paper. Scaling prediction can therefore help prioritize architectures, optimize resource allocation, and reduce experimentation cost. Recent studies have demonstrated that such predictions can be surprisingly accurate in practice (Ruan et al., 2025; Chen et al., 2024; Zhang et al., 2024; Wu & Tang, 2024; Xu et al., 2025), reinforcing the value of scaling laws as an essential tool for efficient LLM development.

While promising, scaling prediction faces two fundamental challenges that remain underexplored. First, there is no guarantee on the accuracy or confidence of the predicted results. Existing methods primarily employ regression fitting with small-scale model quantities and extrapolate performances to large models. However, scaling prediction is intrinsically an *out-of-distribution* (OOD) prediction problem Liu et al. (2021); Ye et al. (2021) (i.e., using data from small-scale regime to predict performance in large-scale regimes) whose optimal learning requires very careful treatment. Second, as we found, existing approaches of simply running regression on random samples suffer from inherent instability due to its nature of extrapolation from small models to the large. Due to the test with

OOD samples, small variations or noise in the observed data can lead to disproportionately large uncertainty in the predictions (Hendrycks et al., 2021; Liu et al., 2021).

In this work, we make three contributions to address these challenges. First, we introduce a metric coined *Equivalent Sample Size* (ESS) to quantify the quality of scaling prediction. ESS provides an intuitive interpretation: it represents how many test examples would have been required via direct, In-Distribution (ID) evaluation to achieve the same accuracy and confidence as that of the scaling prediction. This unified quantification allows practitioners to compare prediction quality across different experimental setups using different model sizes. Second, we conduct a systematic study of the scaling prediction uncertainty induced by extrapolation and solve for the optimal design of the sample regime (e.g., model sizes) that achieves the maximum ESS. Specifically, under computational budget constraints, we develop a polynomial-time algorithm to determine which (smaller) model sizes to evaluate to minimize prediction uncertainty, thereby maximizing ESS. This design ensures that limited evaluation resources are used most effectively, enabling practitioners to decide whether training additional small models is worthwhile to improve prediction quality. Third, we illustrate the use of ESS and our algorithm by predicting model emergent capabilities. We find that using only three models from the OPT family is sufficient to produce a prediction on a large target model with accuracy and ESS comparable to that of using all seven available small models, leading to a significant reduction in the compute cost. The ESS also reflects the prediction quality, thus delivering reliable LLM performance predictions.

Taken together, these contributions offer a principled foundation for predictive evaluation of LLMs via scaling prediction. By introducing ESS as a measure of prediction quality and developing algorithms for optimal evaluation design, *our framework leads to both more efficient scaling prediction via optimized sample choice and more reliable predictions that are crucial for real-world deployment.* We hope this framework will encourage both researchers and practitioners to *rethink scaling law not as a passive learning task but as a strategic resource allocation problem* that can be optimized to balance accuracy, cost, and confidence.

The rest of this paper is organized as follows. Section 2 reviews the related literature and Section 3 formulates the problem. Section 4 introduces ESS for uncertainty quantification. Section 5 reveals that extrapolation inherently induces a high variance of scaling prediction, whiles Section 6 proposes an algorithm to find the optimal design for fitting the scaling law. Section 7 includes experiment results. Conclusion and further discussion are included in Section 8.

## 2 RELATED WORK

Scaling laws have emerged as a powerful tool for understanding and predicting the behavior of LLMs. They reveal a consistent power-law relationship between an LLM's pre-training loss or downstream-task performance and its design factors, particularly compute measures such as training FLOPs, dataset size, and model parameters (Cortes et al., 1993; Kaplan et al., 2020; Brown et al., 2020; Hoffmann et al., 2022; Zhai et al., 2022; Bahri et al., 2024; Gadre et al., 2024). Early work establishes these patterns empirically, e.g., Rosenfeld et al. (2019); Kaplan et al. (2020) shows that increasing a single dimension can lead to a smaller training loss; Hoffmann et al. (2022) proposed the Chinchilla laws to balance model size and data tokens for a fixed compute budget. More recent efforts refines these insights and explore efficient and accurate scaling laws: Alabdulmohsin et al. (2022); Caballero et al. (2022) extended beyond power-laws to more flexible functional forms; Gadre et al. (2024); Owen (2024) studied scaling on downstream tasks; Polo et al. (2024); Ruan et al. (2025) proposed scaling across model families; Choshen et al. (2024) proposed to reduce the training cost of scaling laws by choosing models sizes and tokens, while Hägele et al. (2024) advocated training models with a constant learning rate. These relationships enable researchers to forecast the performance of larger, more expensive models by leveraging empirical observations from smaller ones, thereby avoiding the prohibitive costs of direct training and evaluation.

Existing scaling prediction methods mainly fall into two categories. The first category directly fits an end-to-end scaling law, where model performance is expressed as a power-law function of compute measures. Using observations from smaller models, the fitted curve is then extrapolated to predict the behavior of larger target models (Wu & Tang, 2024; Du et al., 2024). This approach is simple and popular due to its interpretability. However, its reliance on a single functional form (typically log-linear) makes it highly sensitive to deviations from the assumed power law.

The second category introduces an intermediate quantity that itself scales with compute and serves as a bridge between raw resources and final performance. Examples of such intermediate quantity include pre-training loss (Chen et al., 2024) and model capability scores (Ruan et al., 2025; Polo et al., 2024). Once this intermediate variable is estimated via scaling laws, researchers then model its relationship with downstream-task accuracy using flexible predictors such as logistic regression (Xu et al., 2025; Ruan et al., 2025) or neural networks (Ye et al., 2023; Zhang et al., 2024). This two-stage approach often improves prediction accuracy, as the intermediate quantity captures generalizable patterns across tasks or model families.

In both approaches, the key step is extrapolating relationships fitted on small models into much larger, unseen regimes. Such extrapolation is fundamentally unstable: while small estimation errors may be tolerable in-distribution, they are amplified dramatically when extended out-of-distribution to trillion-parameter LLMs. This limitation, further analyzed in Section 5, motivates the need for systematic uncertainty quantification to make scaling prediction reliable for guiding the future LLM evaluations.

On the technical side, our work is related to out-of-distribution (OOD) generalization. For a comprehensive review of OOD literature, we refer to Liu et al. (2021). Generally speaking, controlling OOD generalization is fundamentally difficult, as the test data regime is unseen in the training data. Common approaches typically assume certain relationship between OOD domain and training domain, such as causal learning (Peters et al., 2016), invariant learning (Arjovsky et al., 2019; Zhao et al., 2019), and meta learning (Li et al., 2018). In contrast to these general OOD frameworks, scaling prediction owns a unique structure of extrapolating from small size to large size through power laws, hence allowing more efficient and tractable solutions.

## 3   THE SCALING PREDICTION PROBLEM AND SOURCES OF UNCERTAINTY

This section reviews the process of scaling prediction of model performance and demonstrates that such prediction can be highly uncertain and hence unreliable, which is overlooked in this field.

**Background.** Suppose we have evaluated the performance of some small models $\{f_1, \ldots, f_M\}$ on some tasks $\{T_1, \ldots, T_K\}$, denoted as $P_{m,t}, m = 1, \ldots, M, \ t = 1, \ldots, K$. Without loss of generality, we take $P \in (0, 1)$, as any metric can be monotonically mapped to this range. Our goal is to predict the performance of a large model $f^*$ from the same family on a task $T \in T_1, \ldots, T_K$.

**Scaling Prediction Process.** We unify both end-to-end and intermediate scaling-law approaches as the following process. *Step 1:* Extract a critical quantity (e.g., the capability score or model performance) of training models as $Y_1, \ldots, Y_M$. For notation simplicity, we assume that the critical quantity $Y$ is a scalar, as the vector scenario can be analyzed in an analogous manner coordinate-wise. *Step 2:* Fit a power-law model such that

$$Y = \alpha + \beta X + \epsilon, \tag{1}$$

where $X \in \mathbb{R}^p$ encodes design factors such as the logarithm of number of parameters, size of training data, and FLOPs, $\epsilon$ is Gaussian noise, and $\alpha \in \mathbb{R}, \beta \in \mathbb{R}^p$ are coefficients. Eq. (1) explains the name of scaling prediction, as one extrapolates the critical quantity $Y$ from the small-model regime to larger $X$. *Step 3:* Translate the critical quantity to model performance $P$ as follows:

$$P = \sigma(\omega Y + b), \tag{2}$$

where $\sigma(\cdot)$ is a monotone link function, e.g., $\sigma(z) = 1/(1+e^{-z})$ corresponds to a logistic regression model, and $\omega, b$ are coefficients. End-to-end scaling laws are recovered by taking $\sigma(z) = z$, $\omega = 1$, and $b = 0$. We discuss how our framework can be extended to broader and more realistic scenarios by relaxing assumptions such as linearity and Gaussian noise in Appendix E.3.

*Remark* 1 (Unversality of the Link Function). A nature principle in scaling prediction is that, while the critical quantity $Y$ can be family-specific, the relationship between $Y$ and the final performance $P$ is largely universal across families. For example, a model's accuracy on math problems depends mostly on its underlying math capability, regardless of architecture or training dynamics; however, the rate at which this capability grows with model size varies from family to family. Consequently, the link function in Eq. (2) can often be well-estimated by leveraging performance data from other model families (Chen et al., 2024; Ruan et al., 2025).

**Prediction Uncertainty.** The final predictor $\widehat{P}$ involves two main sources of uncertainty. The first stems from intrinsic random noise in the training data, represented by $\epsilon$ in Eq. (1). This noise arises from measurement error and and randomness in the model training and lies largely beyond the practitioner's control. Nevertheless, this source can be accurately estimated given a well-specified scaling model Eq.s (1) and (2).

As such, this work will focus on the second source: uncertainty introduced by extrapolation. Scaling prediction fits Eq. (1) on small models and use it to predict the performance of much larger ones. Because this applies patterns learned in a limited regime to an unseen region, even slight noise or variation in the observed data can lead to disproportionately large uncertainty in the predictions.

To our knowledge, the uncertainty of scaling prediction has not been rigorously quantified in prior work. We therefore propose a framework to quantify and reduce the uncertainty inherent to extrapolation in the following sections.

*Remark* 2 (Prediction Correctness v.s. Confidence). Under the well-specified model assumption used in the main paper, high prediction confidence reliably indicates high prediction correctness. However, when the model is mis-specified, the prediction may become biased, leading to situations where the model is "confidently wrong." In Appendix E.3, we discuss how the Equivalent Sample Size (ESS) can be used as a diagnostic tool to detect such mis-specification.

## 4    UNCERTAINTY QUANTIFICATION VIA EQUIVALENT SAMPLE SIZE

In this section, we address the problem of quantifying the reliability of scaling prediction. We introduce a measure called *Equivalent Sample Size (ESS)*, which has a natural interpretation from a cost–benefit perspective. Intuitively, ESS compares the information gained by fitting a scaling law on smaller models to that obtained from directly evaluating the target model on an in-distribution test dataset. It represents the number of test examples one would need in direct evaluation to achieve the same level of accuracy and confidence as the scaling prediction.

**Motivation of ESS.** Suppose that we want to evaluate the performance $P$ of a target model $f^*$. Following Eq.s (1) and (2), scaling prediction induces a probability distribution over $P$. Alternatively, one could evaluate $f$ on a test dataset. Given $n$ samples $(S_i, R_i)$, $i = 1, \ldots, n$, where $S_i$ are prompts and $R_i$ are expected responses, the empirical performance is $\widehat{P}_n = n^{-1} \sum_{i=1}^n \ell(f(S_i), R_i)$, where $\ell$ is a loss function. For any bounded loss function $\ell$, a valid $(1 - \delta)$ confidence interval for $\widehat{P}_n$ can be derived by applying Hoeffding's inequality (Hoeffding, 1963) to the random variables $\ell(f(S_i), R_i)$. For example, under the zero-one loss, it yields $[\widehat{P}_n - \epsilon_n, \widehat{P}_n + \epsilon_n]$, where $\epsilon_n = \sqrt{\ln(1/\delta)/(2n)}$. Clearly, these two approaches achieve comparable accuracy if their CIs have the same length. Notably, confidence interval length alone does not capture the difficulty of evaluation, since the same interval width may require dramatically different numbers of test samples depending on the underlying variance. In contrast, ESS can faithfully quantify this difficulty by incorporating the full predictive distribution, thereby providing a principled measure of the quality and practical value of scaling prediction. We elaborate this point in Appendix B.

We formulate this idea as follows.

**Definition 1** (Equivalent Sample Size). Let $\widehat{P}_n$ and $\widetilde{P}$ denote the predictive distributions from direct evaluation and scaling prediction, respectively. Let $\widehat{D}_n(\delta)$ and $\widetilde{D}(\delta)$ be the minimal lengths of their $(1 - \delta)$ confidence intervals. We say that $\widetilde{P}$ has $(n, \delta)$-*equivalent sample size* if $n$ satisfies $\widehat{D}_n(\delta) = \widetilde{D}(\delta)$. As a special case, we have $\widehat{D}_n(\delta) = \sqrt{2\ln(1/\delta)/n}$ when $\widehat{P}_n$ is the empirical average.

The interpretation is straightforward: the scaling prediction achieves the same accuracy as directly testing the target model on $n$ test points. In what follows, we fix $\delta = 0.05$ and refer to this quantity simply as the *effective sample size* unless otherwise noted.

**Practical Implications of ESS.** ESS provides a principled way to compare scaling prediction with direct evaluation under a fixed compute budget. A practitioner can either (i) allocate resources to directly test $f^*$ on $n$ samples, or (ii) evaluate a set of smaller models and fit a scaling law. ESS quantifies the trade-off: if the ESS exceeds $n$, scaling prediction delivers higher accuracy per unit cost. Based on this, the next section further explores how to select the number of models and their

sizes to optimally learn the scaling law and improve efficiency. In addition, ESS can be estimated before any large-scale evaluation, enabling informed decisions in advance.

**Connection to Variance of Critical Quantity.** ESS is tightly linked to the uncertainty in predicting the critical quantity $Y$. In particular, a smaller variance of $Y$ leads to a larger ESS, as formalized in Proposition 4.1. This connection highlights that controlling the variance of $Y$ is key to improving the reliability of scaling predictions. Accordingly, the following sections focus on analyzing and minimizing $\mathrm{var}(Y)$. The complete proof of all propositions and theorems throughout the paper are included in Appendix A.

**Proposition 4.1.** *When the parameters of Eq. (2) are fixed, ESS increases monotonically as the variance of the critical quantity $Y$ decreases.*

## 5 EXTRAPOLATION AMPLIFIES PREDICTION UNCERTAINTY

We now analyze how extrapolation inflates the variance of scaling predictions and, by Proposition 4.1, reduces ESS. For the illustration purpose, we use the logarithm of model size as the design factor $X$, with $X_*$ corresponding to the target model $f^*$. Without loss of generality we assume $X \in [0, \infty)$; otherwise, we can shift and rescale $X$ to ensure non-negative.

We introduce the following notations before deriving $\mathrm{var}(Y)$, the key to prediction uncertainty. Recall the scaling model (1), we denote the variance of the noise $\epsilon$ as $\sigma^2$. For $M$ training models, define sample means $\overline{X}_M := \frac{1}{M} \sum_{i=1}^{M} X_i$ and the empirical variance $\overline{\sigma}_M^2 := \frac{1}{M} \sum_{i=1}^{M} (X_i - \overline{X}_M)^2$.

**Proposition 5.1** (Variance Characterization of Scaling Prediction). *The variance of the critical quantity obtained by sacling prediction model* (1) *is*

$$\mathrm{var}(\widehat{Y}_*) = \frac{\sigma^2}{M} \cdot \frac{(X_* - \overline{X}_M)^2 + \overline{\sigma}_M^2}{\overline{\sigma}_M^2}. \tag{3}$$

Here, the factor $\sigma^2/M$ reflects the intrinsic random noise in the training data, while $(X_* - \overline{X}_M)^2/\overline{\sigma}_M^2$ captures how far the target model lies outside the training range. In an extrapolation setting, $X_*$ is typically much larger than $\overline{X}_M$, so this term dominates the intrinsic noise and leads to a large variance. In contrast, in classical interpolation where $X_* \leq \max_i X_i$, we have $(X_* - \overline{X}_M)^2/\overline{\sigma}_M^2 \leq 1$, keeping the variance comparable to the intrinsic noise level. This difference highlights the inherent instability of scaling predictions, which necessarily extrapolate to larger models with $X_* > \max_i X_i$.

**Example 1.** *Suppose a model with one million parameters corresponds to $X = 0$, and $X_i$'s follow IID exponential distribution $Exp(\lambda)$ so that $\mathbb{P}(X_i = x) = \lambda e^{-\lambda x}, x \geq 0$. For a moderate or large $M$, we have $\mathrm{var}(\widehat{Y}_*) \approx M^{-1}\sigma^2\{1 + (\lambda X_* - 1)^2\}$, since $\overline{X}_M \approx \mathbb{E}(X) = 1/\lambda$ and $\overline{\sigma}_M^2 \approx \mathrm{var}(X) = 1/\lambda^2$. When $\lambda = 1$, predicting a model of 1,000 billion parameters ($X_* = 6$) yields $\mathrm{var}(\widehat{Y}_*) = 26\sigma^2/M$. In contrast, predicting an in-distribution model ($X_* < \mathbb{E}(X) = 1$) gives at most $2\sigma^2/M$. Thus, extrapolation inflates the variance roughly by a factor of $(X_* - 1)^2$.*

Although derived under a linear regression model (1), this variance amplification phenomenon extends to a broad class of machine learning models, including polynomial regression, $k$-nearest neighbors, and tree-based methods. These estimators face the same challenge: predicting far beyond the observed range leaves few, if any, data points near $X_*$, inevitably increasing the variance of $\widehat{Y}_*$. In response, the next section develops a theory to find the optimal training design to reduce this uncertainty and thereby improve the accuracy of scaling predictions.

## 6 UNCERTAINTY REDUCTION BY ACTIVE SELECTION

The variance bound in Eq. (3) shows that the distribution of design factors for the small models largely determines prediction uncertainty. Moreover, Eq. (3) reveals that this variance can be reduced by (1) increasing the number of training points $M$, (2) evaluating models with larger $X_i$, or (3) increasing the spread of the $X_i$, i.e., increasing their variance. However, evaluating more or larger models quickly becomes prohibitively expensive. We therefore propose *active selection:*

*optimally allocating the compute budget across both the number and the sizes of the smaller models to minimize prediction variance.*

**Objective Function.** Formally, we consider a general problem of predicting the performance of any target model with design factor $X_* \in [x_l, x_u]$. Let $W(x)$ denote the importance weight for each target scale $x$, and $c(x)$ the cost of evaluating a model of size $x$. Beyond the $M$ existing models, suppose we can spend a total compute budget $C$ to evaluate $k$ additional models with factors $X_{M+1}, \ldots, X_{M+k}$. Our goal is to choose $k$ and these new $X_{M_j}$'s to minimize the following weighted prediction variance:

$$\min_{k \text{ and } X_{M+j}, j=1,\ldots,k} R(k, X_{M+1}, \ldots, X_{M+k}; X_1, \ldots, X_M, x_l, x_u)$$

$$:= \int_{[x_l, x_u]} \text{var}(\widehat{Y}_*) dW(X_*) \tag{4}$$

$$\text{s.t.} \sum_{j=1}^{k} c(X_{M+j}) \le C, \quad X_{M+j} \ge 0, j = 1, \ldots, k,$$

where $\text{var}(\widehat{Y}_*)$ is the prediction variance given all $M + k$ training points. The special case $x_l = x_u$ recovers the single-target scenario where Eq. (4) reduces to Eq. (3). We also allow $M = 0$, where no prior evaluations exist and the entire learning trajectory must be designed from scratch. In this scenario, one selects the sizes of the small models to create the performance scaling law itself, an idea that motivates the title of our work.

**Optimal Solution.** Solving Eq. (4) is highly non-trivial, as it is a non-convex constrained optimization problem, a class that is typically NP-hard (Benson, 2006b;a). Interestingly, the objective (4) has a special structure: it can be expressed as the ratio of two quadratic functions of $X_i$'s. Exploiting this structure, we derive a key property of the optimal solution that significantly simplify the optimization. Specifically, we show that the optimal design turns out to always evaluate at most three different model scales, though each chosen scale may be sampled multiple times.

Before presenting the main result, we introduce a natural assumption. Without loss of generality, let $X_{M+1} \le X_{M+2} \le \cdots \le X_{M+k}$.

**Assumption 1.** The cost function $c(x)$ and its second order derivative are non-negative and monotonically increasing, i.e., $c(\alpha) > c(\beta) \ge 0$ and $c''(\alpha) > c''(\beta) \ge 0$ for all $\alpha > \beta \ge 0$.

Assumption 1 captures the practical reality that evaluation becomes rapidly more expensive as model size grows. For example, when $x$ is the logarithm of model size, a cost that grows linearly or quadratically in size can be written as $c(x) = e^{a+bx}$ for some constants $a \in \mathbb{R}, b > 0$. It can be verified that Assumption 1 holds for such cost functions.

**Theorem 6.1.** *Under Assumption 1, optimal learning of the scaling law under a given computation budget needs not to use more than two non-zero model scales. Formally, the optimal solution of Eq. (4) must exhibit one of the following two properties:*

*(1)* $0 = X_{M+1}^* = \cdots = X_{M+k_1}^* < X_{M+k_1+1}^* = \cdots X_{M+k_2}^* < X_{M+k_2+1}^* \cdots = X_{M+k}^*$, *and* $\sum_{j=1}^{k} c(X_{M+k}^*) = C$, *where* $k_1, k_2$ *are non-negative integers such that* $0 \le k_1 < k_2 < k$; *or*

*(2)* $0 = X_{M+1}^* = \cdots = X_{M+k_1}^* < X_{M+k_1+1}^* = \cdots = X_{M+k}^*$.

We illustrate the intuition behind Theorem 6.1 by examining the special case of predicting a single target model. In this setting, optimizing (4) reduces to minimizing (3) under a convex budget constraint. A natural strategy is to fix the number of additional models $k$ and then determine their optimal sizes. The key step is to *"break" the ratio structure* in (3) by conditioning on the average model size $\overline{X}_{M+k}$. This renders the numerator constant, so the problem reduces to maximizing the empirical variance. Using variational analysis, we show that any optimal solution contains at most two non-zero scales; otherwise, reallocating budget from mid-size to larger models would yield a smaller variance.

*Remark* 3 (Budget Need Not Be Exhausted). Theorem 6.1 implies that the budget is fully used only when the optimal solution involves two nonzero scales. Notably, when the budget is small, it can be optimal to leave part of it unused. Intuitively, with a small budget it is better to reduce variance by

replicating small models rather than paying for larger ones, so the optimal design may deliberately underspend. For instance, suppose we already have four models of sizes $0.5, 1, 1.5, 2$, and a cost function $c(x) = 0.3e^x$. If the budget is $C = 1$, the optimal solution adds three new models of size $0$ for a total cost of $0.9 < C$. Increasing the budget to $C = 3$ changes the optimal solution to four models of size $0$ plus one medium-scale model of size $1.8$.

**Implication on Efficient Optimization.** Theorem 6.1 not only reveals an interesting property of the optimal solution, but also hints at an efficient polynomial time algorithm for solving the non-convex problem (4) since the characterization of the optimal solution helps drastically reduce the search space. In particular, if the optimal solution only features two model scales, we only need to optimize the single nonzero scale. For an optimal solution with two nonzero scales, the additional constraint $\sum_{j=1}^{k} c(X_{M+j}^*) = C$ again reduces the problem to a single-variable search. Thus, solving (4) reduces to enumerating all feasible triples $0 \leq k_1 \leq k_2 \leq k \leq C/c(0)$ and optimizing a one-dimensional Lipschitz-continuous objective. Pseudo-code of this procedure is in Appendix C.

**How ESS and Active Selection Guide LLM Pre-training Decisions.** The ESS characterizes uncertainty in scaling prediction. A small ESS indicates high variance in the predicted results, suggesting that the current scaling curve lacks stability. In such cases, pre-training teams are advised to gather additional data from smaller models before committing to the costly training of a large model. Conversely, a large ESS signifies high confidence in the fitted scaling relationship, enabling teams to make informed decisions about resource investment.

In addition, Theorem 6.1 offers theoretical guidance for proactively planning the architecture and size of future models within an LLM family. Specifically, it establishes the optimal model size allocation strategy that maximizes the ESS under compute constraints, minimizing prediction uncertainty by Proposition 4.1. Theorem 6.1 thus helps pre-training teams design more effective exploration campaigns, accelerating the development of high-performing models with reduced risk.

# 7 EXPERIMENTS

We evaluate our approach on both simulated and real-world data to show that (1) Equivalent Sample Size (ESS) effectively quantifies and interprets scaling prediction uncertainty, and (2) our active selection algorithm reduces prediction uncertainty under a fixed cost budget, providing practical guidance for optimal experimental design and thus facilitating the learning of performance scaling laws. Additional experiments and full details are provided in Appendix D.

## 7.1 SIMULATED STUDY

Our first experiment predicts the performance of target models with size $X_* \in (4, 7)$. Following Ruan et al. (2025), we focus on the LLAMA-2 family and its emergent capabilities (Srivastava et al., 2023; Wei et al., 2022) on four tasks: word unscramble, Persian QA, 3-digit subtraction, and 2-digit multiplication.

**Data Generation.** For illustration, consider the LLAMA-2 family on the word-unscramble task. We take the coefficients of scaling models (1) and (2) from Ruan et al. (2025). The critical quantity and model size (in logarithm) follow $Y = 0.52X - 0.9 + \epsilon$, where $\epsilon$ is a standard Gaussian noise with standard deviation equal to $0.2$. Model performance $P$ is linked to $Y$ by $P = \sigma(2Y - 6.11)$, where $\sigma(z) = 1/(1 + e^{-z})$ is the standard logistic function. The evaluation cost grows with size as $C(x) = 0.3e^x$.

**Methods.** We compare two methods, Base and Optimal, in terms of their predicted model performance measured by ESS. Base represents the standard practice of fitting the scaling law in (1) using training models whose sizes are not deliberately chosen (Chen et al., 2024; Ruan et al., 2025; Xu et al., 2025). In our experiment, Base samples $M = 10$ models to fit the scaling law, which are IID from a truncated exponential distribution $truncExp(\lambda = 1, b = 3)$, where $b$ is the upper bound of the truncation.

Subject to the same total evaluation cost as Base, Optimal employs our active selection algorithm to choose model sizes that minimize prediction variance, and thereby maximize ESS. We expect Op-

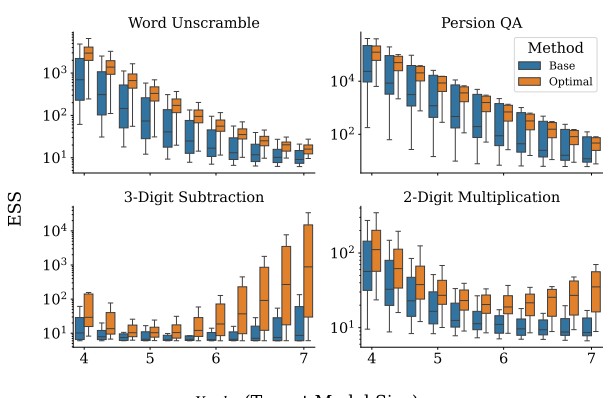

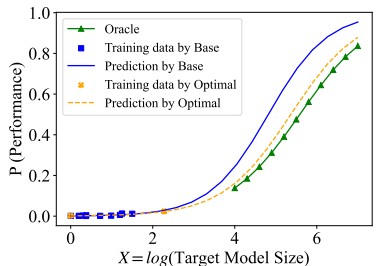

Figure 1: Box-plot of the equivalent sample size (ESS) at varied model size by classic scaling-law-based approach ('Base') and our proposed adaptive selection algorithm ('Optimal') under the same budget. ESS of prediction by models selected by Optimal is significantly higher than that by Base, indicating that Optimal efficiently allocates the budget for maximally improving the reliability of scaling prediction.

Figure 2: A typical realization of training points and predicted performance curves by classic scaling-law-based approach ('Base') and our proposed adaptive selection algorithm ('Optimal'). 'Oracle' is the true model performance. Optimal stabilizes the curve fit and improves prediction by strategically selecting a few key models that expanding the size range.

timal to achieve substantially higher ESS than Base because it strategically allocates the evaluation budget to reduce the uncertainty inherent in scaling predictions.

**Results and Findings.** We ran 20 replicates and present a box-plot of ESS versus target model size in Figure 1. The figure shows that Optimal consistently achieves a much higher ESS than Base, especially for large models. For example, in the word unscramble task, the median ESS at $X = 4$ increases from about 500 (Base) to 3000 (Optimal), which is a six-fold gain. It means that the prediction error and uncertainty are significantly reduced by properly selecting training models, aligning with our theoretical findings.

Figure 2, which depicts a typical simulation run, illustrates the reason behind: Base samples mostly small models (blue squares), whereas Optimal strategically selects a few key models (yellow crosses, two around size 2.2 and four near size 0) to expand the size range and stabilize the scaling-law fit.

Moreover, ESS effectively reflects the scaling prediction quality. As shown in Figure 1, ESS remains very low for the 2-Digit Multiplication task, equivalent to only a few dozen test points, indicating that the prediction can be inaccurate and non-confident. We further confirm this implication of ESS by analyzing the prediction error in Appendix D.

## 7.2 REAL-WORLD APPLICATION: PREDICTING EMERGENT CAPABILITIES

In our simulation studies, the proposed active select algorithm identifies the optimal number and sizes of training models for a given cost budget. In the subsequent real-world experiment, however, model sizes cannot be freely chosen because we cannot access or evaluate new models of arbitrary size. Instead, we must work with the set of models already available to fit the scaling law. Under this constraint, we show that prediction variance can still be reduced, and ESS increased, by applying our active-selection algorithm to choose a subset of the available models that minimizes Eq. (4).

**Dataset Collection.** We evaluate four emergent LLM capabilities of LLMs: word unscramble, Persian QA, 3-digit subtraction and 2-digit multiplication. We use 72 publicly available models drawn from families such as LLaMA2, Qwen1.5, Falcon, GPT-Neo, OPT, Bloom, and Pythia. Their benchmark performance on datasets including MMLU, HellaSwag, GSM8K, and HumanEval is used to extract the critical quantity $Y$, following Ruan et al. (2025). We also collect their ground-truth performance on the four emergent tasks and estimate the link function in Eq. (2).

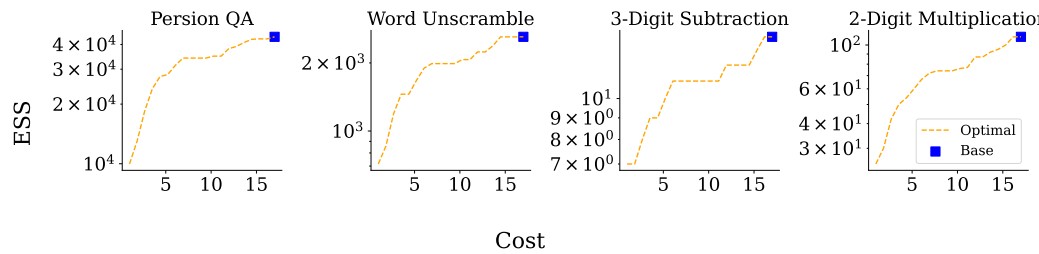

Figure 3: ESS by training on a subset of small OPT models selected by Optimal within a cost budget. Base uses all seven models. Optimal achieves a comparable ESS to Base under a significantly smaller cost.

For the target family, we choose OPT because it offers sufficient models to fit a performance scaling law. Specifically, it has eight models with 125M, 350M, 1.3B, 3B, 7B, 13B, 30B, and 66B parameters. We designate the 66B model as the prediction target, while the remaining seven can be evaluated on the emergent tasks for training. Compute measure $X$ is log-FLOPs calculated as $X = \log(6N \cdot D)$ (Kaplan et al., 2020), where $N$ is the model size and $D = 0.18$ is the pre-training data size. Because all OPT models share the same $D$, $X$ is linearly correlated with model size. The evaluation cost is $c(x) = 0.3e^x$.

**Methods.** We compare the ESS of scaling prediction using the same two methods as in the simulation studies: Base and Optimal. In this real-world experiment, Base fits the power law in (1) using all seven smaller OPT models, while Optimal applies our active selection algorithm to choose a cost-constrained subset of available models for fitting the power law. In the experiment, the cost budget of Optimal is varied from 1 to 16.25, where 16.25 is the total cost of evaluating all models. Full details of the scaling-law fitting procedure are provided in Appendix D. We anticipate that Optimal will deliver predictions of accuracy comparable to Base while requiring substantially lower evaluation cost.

**Results and Findings.** Figure 3 displays the ESS v.s. cost curve for Optimal and the ESS by Base. We have the following key observations from this result. First, Figure 3 demonstrates that a small, well-chosen subset can provide scaling predictions comparable to those obtained from the full set. Across all four tasks, halving the cost budget to $C = 8$ lowers ESS by less than 20%, indicating only a minor loss in predictive power.

Second, we find that four models are consistently selected by our algorithm under a cost budget of $C = 8$: one small model (125M) and three large models (3B, 7B, and 13B). Notably, the total evaluation cost of these models is 6.76, which is below the budget, aligning with our theoretical result that the optimal design need not exhaust the budget.

In Figure 4, we compare the scaling predictions obtained by Base with those derived from the four models selected by Optimal. We report the predicted accuracy, its 95% confidence interval, and the ESS for each method. The result confirms our first observation that the scaling predictions based on Optimal closely match those of Base, while requiring substantially lower cost.

Moreover, the ESS values provide a clear indication of prediction reliability: tasks such as 3-Digit Subtraction and 2-Digit Multiplication exhibit extremely low ESS, signaling that their scaling predictions remain uncertain and may require direct evaluation of the target model or the inclusion of additional smaller models to improve stability.

In summary, these findings underscore that computing the ESS and selecting an appropriate subset allow practitioners to substantially reduce evaluation costs while preserving the reliability of scaling-law predictions; or, equivalently, to enhance reliability without increasing cost.

## 8 CONCLUSION AND FURTHER REMARKS

Our work introduces Equivalent Sample Size (ESS) as a principled metric for quantifying the uncertainty of scaling-based predictive evaluation of LLMs. By analyzing variance induced by extrapo-

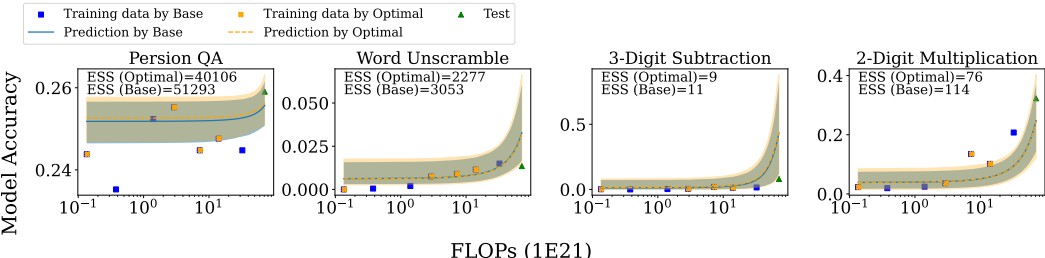

Figure 4: Predicted emergent performance and ESS on the test OPT model. Base uses all seven small OPT models and costs 16.25. Optimal chooses four models to fit the scaling law and costs 6.76. The shaded area is the 95% confidence interval. Optimal achieves nearly the same prediction result and ESS as Base.

lation and proposing an active selection algorithm, we show that practitioners maximize prediction quality by optimally allocating evaluation resources across model sizes.

Despite these contributions, several limitations remain. First, our theoretical analysis assumes that the underlying scaling relationship is correctly specified; if this assumption is violated, predictions may become inaccurate, and the active selection result may be sub-optimal. Developing diagnostics that use ESS to detect model misspecification is therefore a promising direction for future research. Second, our active selection algorithm depends on approximate cost functions and may be sensitive to errors in those estimates, warranting further investigation into how cost-model inaccuracies affect its performance.

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

## A  MISSING PROOFS

**Proof of Proposition 4.1.**

*Proof.* When $\omega$ and $b$ are fixed, any $(1-\delta)$-CI of $Y$, denoted by $[Y_l, Y_u]$, induces a corresponding $(1-\delta)$-CI of $P$, given by $[P_l, P_u]$ where $P_l = \sigma(\omega Y_l + b)$ and $P_u = \sigma(\omega Y_u + b)$. Since the predicted $Y$ is Gaussian, a smaller variance of $Y$ yields a tighter interval $|Y_u - Y_l|$. By the monotonicity of $\sigma(\cdot)$, this translates to a narrower confidence interval $[P_l, P_u]$. According to the definition of ESS, a smaller confidence interval corresponds to a larger ESS, as it reflects the number of test samples needed to achieve the same level of precision through direct evaluation. $\square$

**Proof of Proposition 5.1**

*Proof.* We define $\overline{Y}_M$ and $\overline{XY}_M$, and $\overline{X^2}_M$ analogously to $\overline{X}_M$. By classic learning theory, the least-squares estimates of coefficients in Eq. (1) are

$$\widehat{\alpha}_M = \overline{Y}_M - \widehat{\beta}_M \overline{X}_M, \quad \widehat{\beta}_M = \frac{\overline{XY}_M - \overline{X}_M \overline{Y}_M}{\overline{X^2}_M - \overline{X}_M^2}.$$

Therefore, the predicted critical quantity at $X_*$ is $\widehat{Y}_* = \widehat{\alpha}_M + \widehat{\beta}_M X_*$, whose variance is

$$\mathrm{var}(\widehat{Y}_*) = \mathbb{E}(\widehat{\alpha}_M + \widehat{\beta}_M X_* - \alpha + \beta X_*)^2 = \frac{\sigma^2}{M} \cdot \frac{(X_* - \overline{X}_M)^2 + \overline{\sigma}_M^2}{\overline{\sigma}_M^2}.$$

$\square$

**Proof of Theorem 6.1.**

*Proof.* The original optimization problem can be rewritten as

$$\min_{k \text{ and } X_{M+j}, j=1,\ldots,k} R(k, X_{M+1}, \ldots, X_{M+k}; X_1, \ldots, X_M, x_l, x_u)$$

$$:= \int_{[x_l, x_u]} \frac{\sigma^2}{M+k} \cdot \frac{(x - \overline{X}_{M+k})^2 + \overline{\sigma}_{M+k}^2}{\overline{\sigma}_{M+k}^2} dW(x)$$

$$= \frac{\sigma^2}{(M+k)\overline{\sigma}_{M+k}^2} \cdot \left[ \{\overline{X}_{M+k} - \mathbb{E}_W(X_*)\}^2 + \mathrm{var}_W(X_*) + \overline{\sigma}_{M+k}^2 \right], \quad (5)$$

$$\text{s.t.} \sum_{j=1}^{k} c(X_{M+j}) \le C,$$

$$X_{M+j} \ge 0, j = 1, \ldots, k.$$

where $\mathbb{E}_W(X_*) = \int_{[x_l, x_u]} x \, dW(x)$ and $\mathrm{var}_W(X_*) = \int_{[x_l, x_u]} (x - E_W(X))^2 dW(x)$.

We first show that the optimal design contains at most three different scales.

**Step 1: Conditioning.** We denote $z_j := X_{M+j}, j = 1, \ldots, k$. For any fixed $k$, let $z_j^*, j = 1, \ldots, k$ be any optimal solution and $Z^* = \sum_{i=1}^{k} z_j^*$ be their sum. Consider a variant of Optimization Problem (1) – more specifically, a simplified version – in which both $k$ and $Z^*$ values are pre-given and fixed; hence the goal is simplified to identify the optimal $z_1^*, z_2^* \cdots, z_k^*$ that minimizes the objective of (1) subject to given condition $\sum_{i=1}^{k} z_j = Z^*$. The advantage of considering this simplified version is that the enumerator now becomes a constant, hence the objective is now equivalent to

maximize its denominator. Specifically, given a fixed $k$ and $Z^*$, we have

$$\min_{k;X_{M+1},\ldots,X_{M+k}} R(k,X_{M+1},\ldots,X_{M+k};X_1,\ldots,X_M,x_l,x_u)$$

$$\Leftrightarrow \min_{z_j,j=1,\ldots,k:\sum z_j=Z^*} \frac{\sigma^2}{(M+k)\overline{\sigma}^2_{M+k}} \cdot \left[\{\overline{X}_{M+k} - \mathbb{E}_W(X_*)\}^2 + \operatorname*{var}_W(X_*) + \overline{\sigma}^2_{M+k}\right]$$

$$\Leftrightarrow \min_{z_j,j=1,\ldots,k:\sum z_j=Z^*} \frac{1}{\overline{\sigma}^2_{M+k}} \Leftrightarrow \max_{z_j,j=1,\ldots,k:\sum z_j=Z^*} \overline{\sigma}^2_{M+k}$$

$$\Leftrightarrow \max_{z_j,j=1,\ldots,k:\sum z_j=Z^*} \sum_{i=1}^{M}(X_i - \overline{X}_{M+k})^2 + \sum_{j=1}^{k}(z_j - \overline{X}_{M+k})^2$$

$$\Leftrightarrow \max_{z_j,j=1,\ldots,k:\sum z_j=Z^*} \sum_{j=1}^{k} z_j^2,$$

since $\overline{X}_{M+k} = (\sum_{i=1}^{M} X_i + \sum_{j=1}^{k} z_j)/(M+k)$ is a constant.

As a result, optimizing (5) is equivalent to maximizing the variance of additional points $z_j$'s. The argument above leads to the following re-formulation of OP (5), by conditioning it on the constraint $Z^* = \sum_{j=1}^{k} z_i^*$:

$$\max_{k \text{ and } z_1,\cdots,z_k} \quad \sum_{i=1}^{k} z_i^2$$

$$\text{subject to} \quad \sum_{i=1}^{k} c(z_i) \leq C, \tag{6}$$

$$\sum_{i=1}^{k} z_i = Z^*,$$

$$z_i \geq 0, \forall i = 1,\cdots,k.$$

Notably, despite its simplification, OP (6) is still challenging to solve because its objective is to maximize a convex function which generally is NP-hard. Nevertheless, we show that for any given $Z^*$ and any $k$ value, any optimal solution to OP (6) must be able to be expressed as $z_1^* = z_2^* \cdots = z_{k_1}^* = 0$, $z_{k_1+1}^* = z_{k_1+2}^* \cdots = z_{k_2}^* = \alpha$ and $z_{k_2+1}^* = z_{k_2+2}^* = \cdots z_k^* = \beta$ for some $0 < \alpha < \beta$ and $0 \leq k_1 \leq k_2 \leq k$.

**Step 2: Proving by contradiction.** Suppose the claim above is not true, then there must exist three non-zero $z_i^*$'s with varied values. Without loss of generality, let these three variables be $0 < z_1^* < z_2^* < z_3^*$. Next we argue that, in such cases, there must exist a way to strictly increase the objective value without violating the constraints, hence contradicting their optimality.

Our argument employs the variational methods. Consider new variables $z_1 = z_1^* - \epsilon_1$, $z_2 = z_2^* + \epsilon_2$ and $z_3 = z_3^* - \epsilon_3$ for some arbitrarily small $\epsilon_1,\epsilon_2,\epsilon_3 > 0$ such that

$$-c'(z_1^*)\epsilon_1 + c'(z_2^*)\epsilon_2 - c'(z_3^*)\epsilon_3 = 0 \tag{7}$$

$$-\epsilon_1 + \epsilon_2 - \epsilon_3 = 0 \tag{8}$$

Specifically, these two constraints ensures that the new variables $z_1,z_2,z_3$ remain feasible for OP (6) when $\epsilon_1,\epsilon_2,\epsilon_3$ are arbitrarily small since their relations expressed above guarantee the variation between $z_1,z_2,z_3$ and $z_1^*,z_2^*,z_3^*$ to be 0 for all constraints. However, we argue that the variation of the objective of OP (6) between $z_1,z_2,z_3$ and $z_1^*,z_2^*,z_3^*$ is strictly positive, i.e., $z_1,z_2,z_3$ achieves higher/better objective. Equality (8) implies $\epsilon_2 = \epsilon_1 + \epsilon_3$. Substituting $\epsilon_2$ in Equality (7), we obtain

$$\epsilon_1[c'(z_2^*) - c'(z_1^*)] = \epsilon_3[c'(z_3^*) - c'(z_2^*)]$$

The mean value theorem implies that there exists $u_1 < u_2$ such that $\epsilon_1 c''(u_1)[z_2^* - z_1^*] = \epsilon_3 c''(u_2)[z_3^* - z_2^*]$ By our assumption, all terms are positive and $0 \leq c''(u_1) < c''(u_2)$, hence we must have

$$\epsilon_1[z_2^* - z_1^*] > \epsilon_3[z_3^* - z_2^*].$$

This implies the variation of the objective under $z_1, z_2, z_3$ and $z_1^*, z_2^*, z_3^*$ is strictly positive, i.e.,

$$
\begin{aligned}
\Delta &= -2z_1^*\epsilon_1 + 2z_2^*\epsilon_2 - 2z_3^*\epsilon_3 \\
&= -2z_1^*\epsilon_1 + 2z_2^*(\epsilon_1 + \epsilon_3) - 2z_3^*\epsilon_3 \\
&= 2[z_2^* - z_1^*]\epsilon_1 - 2[z_3^* - z_2^*]\epsilon_3 \\
&> 0.
\end{aligned}
$$

This contradicts the optimality of $z_1^*, z_2^*, z_3^*$, as desired.

Next, we show that the optimal design must satisfy $\sum_{j=1}^{k} c(z_j) = C$ when there are three scales, namely when $k_1 < k_2 < k$. We prove by constructing a contradiction. Suppose $z_j^*, j = 1, \ldots, k$ is an optimal design of OP (6) such that $\sum_{j=1}^{k} c(z_j^*) < C$ and there are three different scales. WLOG, we can let $0 = z_1^* < z_2^* < z_3^*$. Now, let $z_2 = z_2^* - \epsilon$, $z_3 = z_3^* + \epsilon$, and $z_j = z_j^*$ for $j \neq 2, 3$, where $\epsilon$ is a postive constant. Since $\sum_{j=1}^{k} c(z_j^*) < C$, there exists a sufficiently small $\epsilon$ such that $z_i$'s is a valid solution to OP (6). However, the corresponding objective is

$$
\sum_{j=1}^{k} z_j^2 = \sum_{j=1}^{k} (z_j^*)^2 + 2\epsilon(z_3^* - z_2^*) + 2\epsilon^2 > \sum_{j=1}^{k} (z_j^*)^2,
$$

which is a contradiction. We thus completes the proof. $\qquad\square$

## B EXTENSIONS OF ESS

The ESS definition is not tied to Hoeffding-based confidence intervals. Rather, ESS can leverage the *entire predictive distribution* produced by scaling prediction. The Hoeffding bound is distribution-free and conservative, as it does not exploit properties of $f^*$. In this section, we discuss a Bayesian formulation that can yield tighter ESS estimates.

We use zero-one loss as the loss function for the illustration purpose. For direct evaluation, we construct the distribution of $\widehat{P}_n$ through a Bayesian approach. In particular, we assign a uniform distribution as $\widehat{P}_0$, which serves as a non-information prior distribution. After evaluating each test point, we update the posterior distribution $\widehat{P}_n$ by the Bayesian theorem. As a result, the posterior distribution will follow a Beta distribution $Beta(\alpha, \beta)$, where $\alpha$ and $\beta$ can be interpreted as the number of zeros and ones in the evaluation results, respectively. We can therefore match the distribution of $\widetilde{P}$ and $Beta(\alpha, \beta)$ as follows. Let $m_1 = \mathbb{E}(\widetilde{P})$ and $m_2 = \text{var}(\widetilde{P})$. We can solve $\alpha$ and $\beta$ from the following moment equations:

$$
m_1 = \alpha/(\alpha + \beta),
$$
$$
m_2 = \frac{\alpha\beta}{(\alpha + \beta)^2(\alpha + \beta + 1)}.
$$

In particular, we have the close-form solution

$$
\alpha = m_1[\{m_1(1 - m_1)\}/m_2 - 1],
$$
$$
\beta = (1 - m_1)[\{m_1(1 - m_1)\}/m_2 - 1].
$$

Consquently, the scaling prediction has an ESS equals $\widetilde{n}$, where $\widetilde{n} := \alpha + \beta = \{m_1(1-m_1)\}/m_2 - 1$. We note that $\widetilde{n}$ is guaranteed to be non-negative as $m_2 = \text{var}(\widetilde{P}) = \mathbb{E}(\widetilde{P}^2) - \{\mathbb{E}(\widetilde{P})\}^2 \leq m_1 - (m_1)^2$, where the last step is due to $P \in [0, 1]$.

To illustrate the benefit of ESS, consider a model family with very poor accuracy. In this case, direct evaluation requires only a small number of test samples: the observed outputs are consistently incorrect, the empirical variance is low, and the confidence interval quickly becomes narrow. Scaling prediction reflects this by producing a predictive distribution highly concentrated near low performance, which leads to a **small ESS**, thereby accurately capturing the low evaluation cost needed in practice.

Now consider a model family with moderate accuracy (e.g., $\tilde{5}0\%$). Direct evaluation requires many more samples to obtain a confidence interval of the same width. In this scenario, ESS appropriately increases, reflecting the greater sample size needed to reliably estimate performance.

In summary, ESS can faithfully quantify the evaluation difficulty by incorporating the full predictive distribution, thereby providing a principled measure of the quality and practical value of scaling prediction.

## C    PSEUDO-CODE OF THE ACTIVE SELECTION ALGORITHM

The active selection algorithm for solving (4) is summarized in Algorithm 1, with Algorithm 2 serving as a subroutine that computes the optimal solution for each fixed pair $(k_1, k_2)$.

---

**Algorithm 1** Active Selection

---

**Require:** The budget $C$, cost function $c(\cdot)$, and target region $[x_l, x_u]$
1: **for** $k = 1, 2, \ldots$ **do**
2:     Let $R_k, X_1, \ldots, X_k \leftarrow \texttt{BestDesign}(k)$
3:     Stop if the smallest cost of evaluating $k$ models exceeds $C$
4: **end for**
5: Return the design with the smallest $R_k$
**Output:** The optimal design $X_1, \ldots, X_k$.

---

**Algorithm 2** Best Design with a Fixed Number of Models

---

**Require:** The budget $C$, cost function $c(\cdot)$, number of points $k$, and target region $[x_l, x_u]$ and its distribution $W$.
1: **for** $0 \leq k_1 \leq k_2 \leq k$ **do**
2:     Let $X_1, \ldots, X_{k_1} = 0$, $X_{k_2+1}, \ldots, X_k = x$
3:     Let $X_{k_1+1}, \ldots, X_{k_2} = v$, where $v$ is determined by $\sum_{j=1}^{k} c(X_j) = C$.
4:     Solve $x$ that minimizes Eq. (4), and let $R_{k_1, k_2}$ be the minimum
5: **end for**
6: Return the design with the smallest $R_{k_1, k_2}$
**Output:** The optimal design $X_1, \ldots, X_k$.

---

## D    EXPERIMENT DETAILS AND ADDITIONAL RESULTS.

### D.1    SCALING PREDICTION WITH REAL WORLD DATASET

For the Base method, we follow the procedure described in (Ruan et al., 2025). First, we fit the link function in Eq. (2) using the performance of all models **excluding** the target family (LLaMA2) on the emergent benchmark. The link function is specified as

$$\sigma(x) = h + (1 - h)/(1 + e^{-\omega Y - b}),$$

where $h \in [0, 1]$ and $\omega, b \in \mathbb{R}$ are coefficients.

Next, we fit the scaling law in Eq. (1) using the training OPT models. The critical quantity $Y$ for each model is extracted by applying PCA to the imputed performance matrix $B$. Here, each entry of $B$ is the standardized performance of a training model on a benchmark such as MMLU. The final predicted performance is obtained by plugging the extrapolated critical quantity $Y$ of the target model into Eq. (2).

For Optimal, all steps remain the same except that only a subset of OPT models selected by our active selection algorithm (along with models from other families) are included in the training set.

The $(1 - \delta)$-confidence interval (CI) is constructed as follows. We first compute a $(1 - \delta/2)$-CI for $\widehat{Y}$ using Eq. (3). Then, we obtain a $(1 - \delta/2)$-CI for the link function (2) via bootstrapping. Finally, we combine these results through a plug-in procedure to produce the overall $(1 - \delta)$-CI.

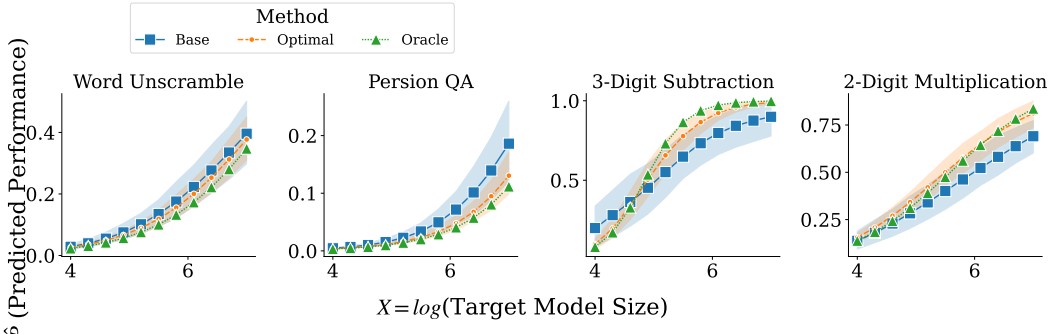

Figure 5: Predicted model performance at varied model size by classic scaling-law-based approach ('Base') and our proposed adaptive selection algorithm ('Optimal'). The true model performance is denoted as 'Oracle'.

## D.2 SIMULATED STUDIES

In addition to ESS, we report the predicted performance of both Base and Optimal in Figure 5. The figure shows that Optimal achieves both lower prediction error and smaller variance than Base, mirroring the ESS improvements observed in Figure 1. This consistency reinforces that ESS reliably captures the quality of scaling predictions and serves as an intuitive measure of their reliability.

## E  RELAXATION OF TECHNICAL ASSUMPTIONS

While our theoretical analysis is built on simplified assumptions to derive clean results, the underlying ideas and optimization pipeline of our framework are general and can be extended to more complex and practical scenarios. Below, we outline several promising directions for such generalizations.

### E.1  RELAXATION OF THE LINEAR MODEL ASSUMPTION

The simple linear model in Eq. (1) serves as a starting point, but our framework is not limited to this form:

- **Multi-variable linear regression.** By allowing a multi-variate input $X$, the model can naturally incorporate multiple explanatory factors, such as model size, training dataset size, learning rate, and other architectural or training hyperparameters.

- **Interaction effects.** Cross-factor terms (e.g., the product of model size and dataset size) can be included in $X$ to capture interaction effects, enabling richer modeling while retaining linearity in parameter space.

- **Nonlinear feature expansion.** Using the kernel trick Hofmann et al. (2008), especially polynomial kernels, we can effectively work in high- or infinite-dimensional feature spaces that include all polynomial combinations of base factors.

- **Approximate linearity.** Many real-world relationships are well approximated by polynomial or piecewise linear functions, making linear modeling a reasonable and practical simplification in many cases.

**Model mis-specification.** In scenarios where the true relationship deviates significantly from our assumed linear form, the predictions may become biased. While this challenge is inherent to all model-based prediction, our framework provides a safeguard: the Equivalent Sample Size (ESS) can be computed on training points to assess the reliability of the model (e.g., using cross validation). A large discrepancy may indicate model misspecification, and how to adaptively refine the model based on such feedback is a promising direction for future work.

### E.2 Relaxation of the Noise Distribution Assumption

Our analysis assumes Gaussian noise primarily for technical convenience. However, the prediction variance in Eq. (3) remains valid under broader settings as long as the noise has bounded variance, which is a standard assumption in learning theory. This implies that the core optimization objective (4) remains largely unaffected. Nevertheless, constructing valid confidence intervals under non-Gaussian noise requires calibration, which can be addressed through methods such as bootstrap resampling or empirical estimation based on the residuals.

### E.3 Relaxation of the Link Function Assumption

We allow the link function $\sigma(\cdot)$ in Eq. (2) to be an arbitrary and possibly unknown function. When $\sigma(\cdot)$ is unknown, non-parametric regression methods such as kernel smoothing or $k$-nearest neighbors can be used to estimate $\sigma(\cdot)$ from the training data. Bootstrapping techniques can then be used to construct prediction intervals, allowing our framework to remain statistically grounded even without strong assumptions on the link function.

## F The Use of Large Language Models Statement

Large language models were used solely as a writing aid. Their use was limited to minor language editing, such as correcting grammar, improving clarity, and polishing the phrasing, without altering the substantive content or analysis of the article.

