# OpenReview forum: "Extrapolating Large Models from the Small: Optimal Learning of Scaling Laws"
_ICLR.cc/2026/Conference — Submitted to ICLR 2026_

### Official Review · Reviewer_JQCo · 2025-10-23

**Soundness:** 1
**Presentation:** 2
**Contribution:** 1
**Rating:** 2
**Confidence:** 4

**Summary:**

This paper addresses the problem of uncertainty in scaling prediction in LLM evaluations and proposes a framework that cuts the evaluation costs when using smaller models to predict the larger model's performance.

**Strengths:**

- The problem of scaling prediction seems to be correctly introduced and motivated in the introduction
- General organization of the paper is in a good state.

**Weaknesses:**

- **W1) Unclear Definition 1.** The definition of the equivalent sample size/ESS (Definition 1) is central to this paper, but the definition is not clear. It does not clearly introduce the "equivalent sample size". The definition just says that a distribution has an equivalent sample size if the number of samples $n$ fulfills a certain condition. Is $n$ the equivalent sample size or what do you mean specifically? If your goal is to introduce a metric for quantifying prediction uncertainty you have to explicitly introduce this metric (this is not sufficiently clear in the current state).  A clear introduction would be especially critical for researchers / practitioners to understand how to adapt your measure for prediction uncertainty.

- **W2) Missing justifications.** The derivation in line 180 suggests that Hoeffding's inequality can be applied, however, applying this inequality requires justification. The problem is also that you do not clearly introduce the random variable for which you want to apply this inequality. Do you want to bound the expectation of the random variable $\ell(f(S_i), R_i)$, where $\ell$ is a loss function? The problem is that we would require this random variable to be bounded in order to apply Hoeffding's inequality (in my understanding), which would require an additional assumption on the loss function. It would also be helpful to cite a source where this inequality is introduced, e.g. (Hoeffding, 1963).

- **W3) General writing-quality is low.** In line 178 $ \hat{P}\_n $ is introduced as the empirical performance, but in Definition 1 it is denoting a distribution (line 184). First $X\_\*$ is a non-negative value (line 212), but later you consider $X_* \in [x_l, x_u]$ (line 254) and it remains unclear why that changes exactly. Section 4 and 5 refer to a Theorem 4.1 but there is no such theorem (just a proposition). $\overline{XY}_M$ in line 216 is never used. The main section 4 aims at quantifying "reliability" instead of uncertainty (line 169) and it remains unclear what is meant with reliability. The paper requires significant polishing.

**Minor weaknesses**
- The paper would generally benefit from a first figure.
- Typo in line 180. Do you mean $\epsilon_n$ instead of $\epsilon$ in the confidence interval?
- The example 1 (lines 231-237) is very loaded with notation and rather challenging to follow.

Overall, the paper requires major revisions throughout the entire manuscript, I cannot recommend it for acceptance in its current state.

**Questions:**

What do we need the "effective sample size" for (line 192)? This is never used again (unless I'm missing something).

Further questions see weaknesses.

---

> ### Author Response · Authors · 2025-11-23
>
> We sincerely thank the reviewer for their time and feedback. In response, we have revised the manuscript to incorporate your suggestions, with major changes highlighted in blue. Below, we summarize our key revisions and clarifications:
>
> **Definition 1.**
>
> We respectfully note that Definition 1 is well-defined. Specifically, we define $\hat{P}_n$ and $\tilde{P}$ as two probability distributions, representing empirical and predicted model performance, respectively. These distributions are not tied to a fixed equation because they are user-specified and may vary across different applications. Once $\hat{P}_n$ and $\tilde{P}$ are given, their corresponding $(1 - \delta)$ confidence intervals $\hat{D}_n(\delta)$ and $\tilde{D}(\delta)$ can be derived using standard techniques. We then define the Equivalent Sample Size (ESS) as the value of $n$ that satisfies the condition $\hat{D}_n(\delta) = \tilde{D}(\delta)$. This definition is general, well-posed, and adaptable to various forms of $\hat{P}_n$ and $\tilde{P}$.
>
> **Justifications for Hoeffding's inequality.**
>
> Thank you for this helpful observation. We have revised the relevant line in the manuscript to improve clarity: "When $\ell$ is zero-one loss, a valid $(1-\delta)$ confidence interval for $\hat{P}_n$ can be derived by applying Hoeffding’s inequality [1] to the random variables $\ell(f(S_i), R_i)$. This yields $[\hat{P}_n-\epsilon_n, \hat{P}_n+\epsilon_n]$, where $\epsilon_n = \sqrt{\ln(1/\delta)/(2n)}$."
>
> [1] Wassily Hoeffding. Probability inequalities for sums of bounded random variables. Journal of the American statistical association, 58(301):13–30, 1963.
>
> **Notation and Writing.**
>
> We respectively argue that the notations in our manuscript is self-contained and consistent:
>
> - $\hat{P}_n$ refers to an empirical distribution derived from direct evaluation. We state in Definition 1 that it can be instantiated as the empirical average over test points.
>
> - $X^\ast$ denotes the design factor of a model and is consistently treated as a non-negative value throughout the paper. In Line 212, we state that we consider evaluating a single model $f^\ast$ with design factor $X^\ast$, and in Line 254 we generalize to any $X^\ast$ within a non-negative interval.
>
> - $\bar{XY}_M$ appears in the proof of Proposition 5.1.
>
>
> **Importance and Role of ESS.**
>
> The Equivalent Sample Size (ESS) provides an interpretable measure of uncertainty in scaling-based predictions. A larger ESS implies higher confidence in the prediction result, while a smaller ESS indicates greater uncertainty. In Proposition 4.1, we formally connect maximizing ESS to minimizing the optimization objective in Eq. (4). Our proposed algorithm in Section 6 is designed to solve this optimization efficiently.
>
> Figures 1 and 3 empirically demonstrate that ESS serves as a useful and interpretable diagnostic for prediction reliability, providing practitioners with actionable insights on when predictions are statistically trustworthy.
>
>
> We hope these revisions and clarifications adequately address your concerns. Please let us know if you have any further questions or suggestions.

---

> ### Comment · Reviewer_JQCo · 2025-11-27
>
> Thank you for your response. I would have a few follow-up questions.
>
> Thank you for clarifying definition 1. I still believe the paper would benefit from writing this more clearly in the paper as well. I'm also concerned regarding the motivation of this definition in the paragraph above definition 1, which is not sufficiently clear to me. The motivation is missing a justification why this equivalent sample size is indeed a good "interpretable" measure of uncertainty. In this context, I'm also following the discussion with other reviewers and share concerns for example with Reviewer CzyY. Would it not be better to report the confidence interval length directly as done usually?
>
> Regarding Hoeffding's inequality, I read your update in the manuscript. However, I still believe applying this inequality requires careful justification that the random variable you consider is bounded. Right now you have only restricted the discussion to the zero-one loss, but does this mean your analysis only works for this particular loss? Does it work with other loss functions commonly used?
>
> Regarding notation, I checked for example the proof of Proposition 5.1 in the Appendix. While I agree that $\bar{XY}_M$ is needed there, it is not clear in the main text for what purpose you introduce these variables, especially if they are only needed in the proof in the Appendix.

---

> > ### Author Response · Authors · 2025-11-27
> >
> > Thank you very much for your continued engagement. We have further updated our manuscript to incorporate your feedback. We summarize the responses below.
> >
> > **Motivation of ESS.**
> >
> > The key motivation behind ESS is that it represents the *amount of direct evaluation* a practitioner would otherwise need to achieve the same level of confidence. In contrast, **confidence interval length alone does not capture the difficulty of evaluation**, since the same interval width may require dramatically different numbers of test samples depending on the target model and task. Importantly, ESS is not tied to Hoeffding-based confidence intervals; rather, it can leverage the *entire predictive distribution* produced by scaling prediction. As discussed in Appendix B, this makes ESS substantially more informative than reporting confidence intervals alone.
> >
> > To illustrate, consider a model family with very poor accuracy. In this case, direct evaluation requires only a small number of test samples: the observed outputs are consistently incorrect, the empirical variance is low, and the confidence interval quickly becomes narrow. Scaling prediction reflects this by producing a predictive distribution highly concentrated near low performance, which leads to a small ESS, thereby accurately capturing the low evaluation cost needed in practice.
> >
> > Now consider a model family with moderate accuracy (e.g., ~50%). Direct evaluation requires many more samples to obtain a confidence interval of the same width. In this scenario, ESS appropriately increases, reflecting the greater sample size needed to reliably estimate performance.
> >
> > In summary, ESS can faithfully quantify the evaluation difficulty by incorporating the full predictive distribution, thereby providing a principled measure of the quality and practical value of scaling prediction.
> >
> > **Hoeffding Inequality.**
> >
> > Thank you for pointing this out. We clarify that Hoeffding’s inequality applies to any bounded loss function. We have revised the relevant sentence as follows: "For any bounded loss function $\ell$, a valid $(1-\delta)$ confidence interval for $\hat{P}_n$ can be derived by applying Hoeffding’s inequality to the random variables $\ell(f(S_i), R_i)$. For example, under the zero-one loss, it yields $[\hat{P}_n-\epsilon_n, \hat{P}_n+\epsilon_n]$, where $\epsilon_n = \sqrt{\ln(1/\delta)/(2n)}$."
> >
> > **Notation.**
> >
> > Following your helpful suggestion, we have moved these definitions to the appendix for improved readability.
> >
> > We hope these clarifications address your concerns fully. Please feel free to let us know if there are any further points we can expand upon.

---

### Official Review · Reviewer_ynTz · 2025-10-31

**Soundness:** 2
**Presentation:** 2
**Contribution:** 1
**Rating:** 2
**Confidence:** 4

**Summary:**

The paper studies using small models to fit scaling laws and then extrapolate to large models to cut evaluation cost, but finds that such extrapolation is out-of-distribution and often unstable. The main contribution of the work is proposing an interpretable ESS metric, explaining the variance-amplification mechanism of out-of-range extrapolation, and providing an optimal selection method for small models.

**Strengths:**

1. The proposed definition and framework are novel and workable.

2. In implementing the algorithm, the objective function and constraints are clearly defined, and the experiments support the theoretical claims.

**Weaknesses:**

1. Reframing the Problem Motivation: The paper compellingly frames the problem around the "prohibitively expensive" cost of evaluation. This is a valid point. However, in the context of SOTA model development, the cost of training is often the dominant bottleneck by several orders of magnitude. The primary industrial use of scaling laws is typically for pre-training decision support (e.g., comparing architectures or data/model allocations) rather than saving post-training evaluation costs. The paper's impact could be significantly strengthened by discussing how the proposed ESS metric and optimal selection algorithm could be adapted for this (arguably more critical) pre-training decision-making scenario.

2. On the 'Model Family' and 'Universal' Link Function Assumption: The framework's practicality hinges on the assumptions made about 'model families.' In practice, scaling laws are often most needed to compare models with subtle but critical differences (e.g., aspect ratios, data mixes), which may already constitute different 'families.' The paper's real-world experiment (Section 7.2) and Remark 1 assume a 'largely universal' link function by fitting it across multiple, distinct model architectures (LLaMA, Qwen, Bloom, etc.). This assumption feels very strong and may not hold in the very scenarios where scaling prediction is most valuable. It would be helpful for the authors to provide more justification for this universality or discuss the sensitivity of their method to this assumption.

3. Reliance on a Correctly Specified Model Form: The paper's core theoretical contributions, such as the variance characterization (Proposition 5.1) and the optimal selection algorithm (Theorem 6.1) , are derived from a specific power-law model (Eq. 1) assuming Gaussian noise. This framework primarily addresses aleatoric uncertainty (noise) and extrapolation variance, assuming the model form itself is correct. As the authors thoughtfully acknowledge in Section 8, this is a key limitation. In practice, model misspecification (i.e., the true scaling relationship is not captured by Eq. 1) is often the largest source of error. This creates a risk that the proposed algorithm could lead to a solution that is "optimally confident" but potentially "incorrect" (i.e., high ESS for a biased prediction). A discussion on how ESS might also help detect or quantify this model form uncertainty would be a valuable addition.

4. Distinguishing Prediction 'Confidence' from 'Correctness': Following the point above, the paper's focus is on minimizing variance to maximize ESS. This is an important and non-trivial contribution. However, it would be beneficial to more clearly delineate this from the challenge of correctness (i.e., bias). The current framework does not seem to penalize a model that is "confidently wrong." It would strengthen the paper to discuss this distinction and whether the optimal selection strategy might inadvertently shift if the goal was to minimize total error (Bias + Variance) rather than just variance.

5. Simplification of Scaling Factors and Interaction Terms: The analysis in Section 5 simplifies the design factor $X$ to a single dimension (log model size) for illustrative purposes. While the model $Y=\alpha+\beta X$ could in principle handle a multi-dimensional $X$, it remains a linear model. This may not be sufficient to capture the complex, non-linear interaction terms between scaling factors [1] have shown to be critical. It is unclear how the optimal selection algorithm (Theorem 6.1) would behave if the true underlying scaling law had such cross-terms.

[1] Houyi Li, Wenzhen Zheng, Qiufeng Wang, Zhenyu Ding, Haoying Wang, Zili Wang, Shijie Xuyang, Ning Ding, Shuigeng Zhou, Xiangyu Zhang, Daxin Jiang“Predictable Scale: Part II, Farseer: A Refined Scaling Law in Large Language Models.” arXiv:2506.10972 [cs.LG]. https://arxiv.org/abs/2506.10972.

**Questions:**

1. Regarding the problem motivation on evaluation cost: Could the authors elaborate on the practical scenarios where this cost is the primary bottleneck, especially in relation to the much larger training costs that scaling laws are typically used to predict? We are interested in how the framework might be repositioned to address the pre-training decision problem.

2. Regarding the 'universal' link function: This assumption is central to the experimental setup . How sensitive is the method to this assumption? In practice, one often needs to compare two very similar, but distinct, model families. How would a practitioner validate whether they can use a shared link function, or how would the method be adapted if they cannot?

3. Regarding the ESS metric (Definition 1) : This provides a valuable measure of confidence based on prediction variance. However, if the underlying scaling model (Eq. 1) is misspecified, ESS might be high for a prediction that is, in fact, highly biased. Could the authors comment on this potential trade-off? Is there a way to use ESS to also help diagnose model misspecification, rather than only quantifying variance?

4. Regarding interaction terms: The theoretical analysis relies on a linear relationship between the critical quantity $Y$ and the design factors $X$ (Eq. 1). How would the theoretical results (change if the true scaling law involved non-linear interaction terms between factors, as suggested by other recent work on scaling laws [e.g., Farseer, Li et al., 2025]?

---

> ### Author Response · Authors · 2025-11-23
>
> We sincerely thank the reviewer for their thoughtful feedback, particularly regarding the reframing of our motivation and the relaxation of technical assumptions. In response, we have revised the manuscript to incorporate your suggestions, with major changes highlighted in blue. Below, we summarize our key revisions and clarifications:
>
> **Reframing the Motivation**
>
> We deeply appreciate your insightful suggestion on better positioning the practical utility of our framework. As also echoed by Reviewer gxSu, our proposed method “possesses considerable practical value—specifically, it can help pre-training teams proactively predict model performance.” We have taken this feedback seriously and revised the introduction to reframe our motivation: providing a principled tool to guide pre-training decisions (e.g., selecting model architectures or parameter sizes) before significant computational investments are made. Additionally, we elaborated in Lines 330–340 on how the Equivalent Sample Size (ESS) and our proposed active model selection algorithm can directly support and optimize pre-training strategies.
>
>
> **Relaxtion of Assumptions (Appendix E).**
>
> Our theoretical results are derived under simplified assumptions to enable clean, interpretable analysis. However, the core ideas and optimization pipeline of our framework are more broadly applicable. In Appendix E of the revised manuscript, we detail how our framework can be generalized to more practical scenarios by relaxing assumptions such as linearity, Gaussian noise, and the specific form of the link function.
>
> In particularly, please find our response to "Reliance on a Correctly Specified Model Form" and "Simplification of Scaling Factors and Interaction Terms" in Appendix E.1.
>
> **Universal Link Function.**
>
> We would like to clarify a possible misunderstanding. The link function $\sigma(\cdot)$ in Eq. (2) models the relationship between a critical quantity $Y$ (not the raw input $X$) and performance $P$, consistent with common scaling law assumptions. For example, a model’s performance on math problems is governed by its underlying mathematical capability $Y$, regardless of architectural differences.
>
> As clarified in Remark 1, models within the same family exhibit principled relationships between their size and critical quantity; the mapping from architecture/training hyperparameters to $Y$ is family-specific, but the link from $Y$ to $P$ is largely shared.
>
> In addition, for cases where the link function is unknown, we discuss in Appendix E.3 how non-parametric methods can estimate it without strong functional assumptions. We have also discussed how to use ESS to detect  potential model mis-specification in Appendix E.1.
>
>
> **Distinguishing Prediction 'Confidence' from 'Correctness'**
>
> Thank you for highlighting this subtle but important distinction. We have added Remark 2 as follows: "Under the well-specified model assumption used in the main paper, high prediction confidence reliably indicates high prediction correctness. However, when the model is mis-specified, the prediction may become biased, leading to situations where the model is "confidently wrong.'' In Appendix E.1, we discuss how the Equivalent Sample Size (ESS) can be used as a diagnostic tool to detect such mis-specification."
>
> We hope our revisions and clarifications adequately address your concerns, and we are happy to clarify or expand on any other questions you may have.

---

### Official Review · Reviewer_CzyY · 2025-11-01

**Soundness:** 3
**Presentation:** 2
**Contribution:** 3
**Rating:** 6
**Confidence:** 4

**Summary:**

This paper addresses the statistical guarantees of the observational scaling law. It introduces a metric called equivalent sample size to quantify prediction uncertainty. The authors also develop an efficient algorithm to maximize ESS within computational budgets. Synthetic and real-world experiments demonstrate the proposed method’s effectiveness.

**Strengths:**

- The paper tackles an important problem regarding formal guarantees for scaling laws.
- The proposed efficient model-selection method is technically sound. Some conclusions are counterintuitive and therefore noteworthy.

**Weaknesses:**

- The proposed framework largely relies on the functional form derived from observational scaling laws. This raises the risk of model misspecification, and the Gaussian assumption may not hold in that case.
- The computation saved by the proposed algorithm seems modest to me in experiments.
 - Minor issue: It would be better to reference the appendix contents in the main paper. Currently, none of the proofs are cited.

**Questions:**

Please refer to those in the weaknesses.

---

> ### Author Response · Authors · 2025-11-23
>
> We sincerely thank the reviewer for recognizing the practical significance of our work and for the thoughtful feedback. In response, we have revised the manuscript to incorporate your suggestions, with major changes highlighted in blue. Below, we summarize our key revisions and clarifications:
>
> **Relaxtion of Assumptions (Appendix E).**
>
> Our theoretical results are derived under simplified assumptions to enable clean, interpretable analysis. However, the core ideas and optimization pipeline of our framework are more broadly applicable. In Appendix E of the revised manuscript, we detail how our framework can be generalized to more practical scenarios by relaxing assumptions such as linearity, Gaussian noise, and model well-specification.
>
> In particular, our results on prediction variance and optimal allocation remain valid as long as the noise has bounded variance. The main difference under non-Gaussian noise lies in calibrating the confidence intervals, which can be handled through resampling techniques such as bootstrapping or empirical residual-based estimation.
>
> Regarding potential model mis-specification, we propose a practical safeguard: the Equivalent Sample Size (ESS) can be computed on the training data (e.g., via cross-validation) to assess the reliability of the fitted scaling law. A substantial deviation in ESS could signal mis-specification. Developing adaptive strategies based on such feedback is an exciting direction for future work.
>
> **Practical Value of Optimal Learning Algorithm.**
>
> As noted by Reviewers gxSu and ynTz, our proposed framework "possesses considerable practical value—specifically, it can help pre-training teams proactively predict model performance." To reflect this perspective more clearly, we have rewritten the introduction to refram our motivation: providing a principled tool that can guide pre-training decisions (e.g., select model architectures or parameter sizes) prior to making large computational investments.
>
> In particular, the value of our theoretical development is supported by the growing practice of using scaling laws to guide large-scale model development. We elaborate on this connection in Lines 330–340 of the revised manuscript.
>
> **References to Appendix Content.**
>
> We have updated the main text to provide clearer references directing readers to relevant appendices for detailed proofs and extended discussions.
>
> We hope our revisions and clarifications adequately address your concerns, and we are happy to clarify or expand on any other questions you may have.

---

> > ### Comment · Reviewer_CzyY · 2025-11-27
> >
> > I appreciate the authors' efforts in the rebuttal. I have a follow-up question: What motivates the Effective Sample Size (ESS), and how does it contribute to the rest of this work? In my opinion, the ESS is determined by the length of the confidence interval. Reporting ESS instead of the confidence interval length offers only minor benefits in terms of interpretability, and the simple math involved can barely be considered a significant contribution of the paper. I wonder if there's anything I might have missed.

---

> > > ### Author Response · Authors · 2025-11-27
> > >
> > > Thank you for reviewing our rebuttal and raising this excellent question.
> > >
> > > The key motivation behind ESS is that it represents the *amount of direct evaluation* a practitioner would otherwise need to achieve the same level of confidence. In contrast, **confidence interval length alone does not capture the difficulty of evaluation**, since the same interval width may require dramatically different numbers of test samples depending on the target model and task. Importantly, ESS is not tied to Hoeffding-based confidence intervals; rather, it can leverage the *entire predictive distribution* produced by scaling prediction. As discussed in Appendix B, this makes ESS substantially more informative than reporting confidence intervals alone.
> > >
> > > To illustrate, consider a model family with very poor accuracy. In this case, direct evaluation requires only a small number of test samples: the observed outputs are consistently incorrect, the empirical variance is low, and the confidence interval quickly becomes narrow. Scaling prediction reflects this by producing a predictive distribution highly concentrated near low performance, which leads to a small ESS, thereby accurately capturing the low evaluation cost needed in practice.
> > >
> > > Now consider a model family with moderate accuracy (e.g., ~50%). Direct evaluation requires many more samples to obtain a confidence interval of the same width. In this scenario, ESS appropriately increases, reflecting the greater sample size needed to reliably estimate performance.
> > >
> > > In summary, ESS can faithfully quantify the evaluation difficulty by incorporating the full predictive distribution, thereby providing a principled measure of the quality and practical value of scaling prediction.
> > >
> > > We hope our responses address your concerns, and we’d be happy to further clarify or expand on any remaining questions.

---

### Official Review · Reviewer_gxSu · 2025-11-09

**Soundness:** 4
**Presentation:** 3
**Contribution:** 2
**Rating:** 6
**Confidence:** 3

**Summary:**

This paper focuses on the confidence intervals of scaling law. It mainly puts forward a series of optimization strategies to address the problem that large model evaluation consumes excessive resources.​

The authors' discussion revolves around two key perspectives:​
1. As the model scales up, the confidence intervals widen;​
2. As the volume of test data increases, the confidence intervals narrow.​

Accordingly, the authors propose that more evaluations should be conducted at critical junctures, with the aim of reducing the widening of confidence intervals during the model scaling process.

**Strengths:**

The authors' discussion angles demonstrate remarkable novelty.

Moreover, they indeed possess considerable practical value—specifically, they can help pre-training teams proactively predict model performance.

**Weaknesses:**

- Some studies have proposed that scaling laws may not follow a simple logarithmic curve—especially with an increase in data repetition. This could lead to biases in the widening of the authors' confidence intervals, greatly undermining the effectiveness of the authors' method.​
- The authors' discussion lacks consideration of learning rates. In fact, most model training processes may involve adjustments to learning rates during multi-stage training, which exerts a significant impact on model performance. Meanwhile, the usability of the authors' method is also compromised.

Chen, Zhengyu, et al. "Revisiting scaling laws for language models: The role of data quality and training strategies." Proceedings of the 63rd Annual Meeting of the Association for Computational Linguistics (Volume 1: Long Papers). 2025.

Muennighoff, Niklas, et al. "Scaling data-constrained language models." Advances in Neural Information Processing Systems 36 (2023): 50358-50376.

Hernandez, Danny, et al. "Scaling laws and interpretability of learning from repeated data." arXiv preprint arXiv:2205.10487 (2022).

**Questions:**

How do the authors plan to mitigate the aforementioned weaknesses?

---

> ### Author Response · Authors · 2025-11-23
>
> We sincerely thank the reviewer for recognizing the practical significance of our work and for the thoughtful feedback. In response, we have revised the manuscript to incorporate your suggestions, with major changes highlighted in blue. Below, we summarize our key revisions and clarifications:
>
> **Relaxtion of Assumptions (Appendix E).**
>
> Our theoretical results are derived under simplified assumptions to enable clean, interpretable analysis. However, the core ideas and optimization pipeline of our framework are more broadly applicable. In Appendix E of the revised manuscript, we detail how our framework can be generalized to more practical scenarios by relaxing assumptions such as linearity, Gaussian noise, and the specific form of the link function.
>
> In particular, we allow the link function between critical quantity $Y$ and model performance $P$ to be arbitrary and possibly unknown, rather than fixed (e.g., logarithmic). In such cases, non-parametric regression methods such as kernel smoothing or $k$-nearest neighbors can be used to estimate the link function. We have also discussed how to use ESS to detect  potential model mis-specification.
>
> **Learning rates.**
>
> We understand the reviewer’s concern regarding the influence of learning rates. As clarified in Remark 1, the premise of scaling laws is that models within the same family exhibit principled relationships between their size and performance. Thus, models trained with dramatically different learning rate schedules may not follow the same scaling trends and can be considered as belonging to different families. Consquently, one practical strategy is to label models trained under different learning rate patterns as distinct families. Our framework and theoretical results can then be applied to a family with specific learning rate schedule, leveraging information from models using other learning rates.
>
>
> We hope our revisions and clarifications adequately address your concerns, and we are happy to clarify or expand on any other questions you may have.

---

### Author Response · Authors · 2025-11-29
**Revision Summary**

Dear Reviewers and Area Chair,

We sincerely appreciate your constructive feedback and the time you’ve dedicated to the review process. Below, we summarize the major revisions and clarifications we have made in response to your comments. These updates are highlighted in blue in the revised manuscript.

1. **Reframed Motivation.** Following the suggestions from Reviewers gxSu and ynTz, we have revised the introduction to better emphasize the practical utility of our framework regarding model training. Specifically, we now highlight how our method can guide critical pre-training decisions (e.g., selecting model architectures or parameter sizes) before committing to costly computational investments.

2. **Relaxtion of Assumptions.** We have added a dedicated section in Appendix E to discuss how our theoretical framework generalizes to more practical settings, addressing the concerns raised by Reviewer gxSu, CzyY, and ynTz. In particular, we relax assumptions such as linearity, Gaussian noise, and model well-specification, and explain how our methods extend to these broader cases.

3. **Expanded Related Work.** The related work section has been substantially expanded to incorporate over ten additional papers, providing a more comprehensive overview of the literature.

4. **Notation Improvements.** We have carefully addressed all notation concerns raised by Reviewer JQCo for improved readability.

We believe these revisions fully address the reviewers’ concerns, and we kindly hope the Area Chair will consider them in their final evaluation. Thank you again for your valuable time!

Sincerely,

Authors of Submission 10071

---

### Meta-Review · Area_Chair_BvLV · 2026-01-06

**Summary:**

While all the reviewers acknowledged the relevance and importance of the proposed work, there are several major concerns about the paper.

1. Reviewers gxSu, CzyY and ynTz all expressed concerns over the strong assumptions on functional form and error distribution.

2. Reviewer gxSu expressed concerns over the learning rates impacting the usability of the model.

3. Reviewer CzyY and ynTz questioned the motivation of the paper: saving evaluation cost does not seem to be the bottleneck of scaling law; the experiments do not show significant cost savings either.

4. Reviewer ynTz questioned the validity of the universal link function assumption and model family assumption.

5. Reviewer ynTz raised potential failure cases where the model is confidently wrong.

6. Reviewer JQCo expressed concerns over the quality of writing, including notations, insufficient discussion of the rationales behind the design of the method.

7. Reviewers CzyY and JQCo had questions about the rationales behind the ESS concept.

8. AC cb2a pointed out that the literature review is sparse.

**Reviewer Concerns:**

1. The authors have added a discussion about releasing the assumptions (Appendix), addressing the concerns over strong assumptions.

2. The authors acknowledged that different learning rates may add challenges to the method, but proposed a tentative solution of labeling different learning rates as different model families.

3. The authors have revised the motivation of the paper as 'providing a principled tool that can guide pre-training decisions'.

4. The authors pointed out that the universal link function assumption is on the relation between $Y$ and $P$, not on $X$ and $P$, which may be a misunderstanding by Reviewer ynTz. However, there is still a lack of discussion on how to construct model families when the design choices in question are very nuanced.

5. The authors acknowledged that the method could fail when the mispcification leads to the method being 'confidently wrong', but proposed a principled method to identify model specification.

6. It turns out that most writing issues suggested by JQCo are misunderstandings.

7. The authors have provided a good explanation of why ESS is unique and informative. I would encourage the authors to add the discussion to the paper.

8. The authors have revised the literature review section.

In short, since the authors decided to change the motivation of the paper to providing a tool to guide pre-training decisions, there would need to be more experiments performed to show the effectiveness of the method for this purpose. However, as suggested by Reviewer ynTz, many design choices are very nuanced (e.g., the learning rate concern raised by Reviewer gxSu), there would need to be stronger empirical evidence to show that the method can distinguish nuanced choices, potentially by properly defining the model families. The current version, therefore, seems a bit premature with the new motivation. I am confident that it will turn into a good paper, given enough time for more experiments supporting the new motivation.

**Reviewer Scores:**

Reviewer gxSu may maintain their score, because they may expect more in-depth analysis on the learning rate.

Reviewer CzyY may raise the score to 8.

Reviewer ynTz may raise the score to 4, and may expect more discussions on model family choices.

Reviewer JQCo may raise the score to 4, because the writing quality can still be improved.

---

### Decision · Program_Chairs · 2026-01-26

Reject